# Mitochondrial Phylogenomics Suggests Complex Evolutionary Pattern of Pronotal Foliaceous Mimicry in Hierodulinae (Mantodea: Mantidae), with Description of a New Species of *Rhombodera* Burmeister, 1838 from China [note 1]

**DOI:** 10.3390/insects13080715

**Published:** 2022-08-09

**Authors:** Qin-Peng Liu, Le Liang, Xin-Yang Zhang, Hao-Kun Li, Chu-Xiang Zhao, Xing-Yue Liu

**Affiliations:** 1Department of Entomology, College of Plant Protection, China Agricultural University, Beijing 100193, China; 2College of Biological Science, China Agricultural University, Beijing 100193, China; 3School of Biomedicine, University of Melbourne, Parkville, VA 3010, Australia; 4College of Software Engineering, Northwestern Polytechnical University, Xi’an 710129, China

**Keywords:** Hierodulinae, mitogenome, phylogeny, foliaceous pronotum, new species, male genitalia

## Abstract

**Simple Summary:**

Rampant morphological homoplasy in Mantodea has confused systematists for a long time. The leaf-like pronotum was long thought to be one of the key characteristics of *Rhombodera* in Hierodulinae. In this study, we describe a new species of *Rhombodera* and discuss the phylogeny as well as complex convergent evolution of leaf-like pronotum in Hierodulinae based on the combination of morphological and mitogenome data. The results reveal the unnatural grouping of the current classification in *Hierodula* and *Rhombodera*, rendering the well-developed lateral expansion invalid as the diagnosis of *Rhombodera*. Mapping the pronotal-foliaceous trait onto our phylogenetic tree show that the origin in multiple lineages of this trait in Hierodulinae may have adaptive significance. The congruent results of the group division between mitochondrial phylogenomics and morphology of male genitalia also highlight the capacity of the use of mitogenomes in resolving the derived mantodean phylogenetic relationship.

**Abstract:**

Hierodulinae is a species-rich mantid subfamily, with some species bearing a notable leaf-like pronotum. However, the evolutionary pattern and taxonomic significance of the leaf-like pronotum are largely unknown. Here, we present a phylogenomic analysis of the Hierodulinae genera *Rhombodera* Burmeister, 1838, and *Hierodula* Burmeister, 1838 based on mitochondrial genomes. We also describe a new species, namely *Rhombodera hyalina* **sp. nov.** from Guangxi, China. Our phylogenetic result, together with the evidence from male genitalia, suggests the division of the Oriental *Hierodula* and *Rhombodera* complex into three clades. We find a complex pattern on the evolution of the leaf-like pronotum, which is present in at least five lineages, respectively, of the above three clades.

## 1. Introduction

Hierodulinae Brunner de Wattenwyl, 1893 is the largest subfamily of Mantidae Latreille, 1802, containing 25 genera and more than 200 species, which widely occur in the Old World [1]. The adults of Hierodulinae show a typical praying mantis body shape with medium to large size, but some species possess a foliate pronotum that is related to leaf-mimicry. 

Historically, Burmeister was one of the early researchers of Hierodulinae, who first proposed two speciose genera, *Rhombodera* Burmeister, 1838, and *Hierodula* Burmeister, 1838 [2]. Brunner de Wattenwyl established a group of four genera into Hierodulae, as the predecessor taxon of Hierodulinae [3]. Giglio-Tos [4,5,6] described a large number of genera and species in the early 1990s, which was followed by some important revisions by Werner and Beier [7,8,9]. However, the comprehensive classification with a clear definition of Hierodulinae refers to a recent work by Schwarz and Roy [10]. 

Southern China has been known as one of the areas with many new taxa of Hierodulinae found in recent five years, such as *Mekongomantis* Schwarz, Ehrmann and Shcherbakov, 2018, *Dracomantis* Shcherbakov and Vermeersch, 2020, *Titanodula menglaensis* Liu, Wang and Yin, 2021, and *Rhombomantis longipennis* Wang et al., 2021. However, *Rhombodera* remains a poorly studied group, which was previously recognized by the width of the pronotum greater than the width of the head and well-developed lateral expansion not enfolding the head anteriorly. It is known that the pronotal leaf-like mimicry is present in many subfamilies of Mantidae, such as Choeradodinae de Saussure, 1869, Tenoderinae Brunner de Wattenwyl, 1893 and Hierodulinae. In Hierodulinae, the foliate pronotum has been found in multiple groups, including *Rhombodera extensicollis* Serville, 1839 in the *Hierodula membranacea* group, *Rhombomantis fusca* Lombardo, 1992 in the *Hierodula patellifera* group, and most species of *Rhombodera s. str.* Burmeister, 1838. Such morphological specialization seems to be a kind of adaptive trait, which, however, might be homoplasious in distantly related species. Previous works have repeatedly pointed out the unnatural grouping of *Hierodula* and *Rhombodera* [10,11,12,13] caused by homoplasy, while few have actually solved related problems. 

Mitogenome has been widely used as a molecular marker of insects for phylogenetic inference at genus or species level and population genetics, given its rapid site evolution, conserved gene composition, and limited recombination [14]. As of 28 June 2022, about 27 complete or nearly complete mitogenomes of Hierodulinae species have been sequenced since the first mitogenome of Mantodea, *Tamolanica tamolana* Brancsik, 1897 was obtained [15]. Mitochondrial phylogenomic studies of Mantodea in the past five years [16,17,18,19,20] have shown generally congruent relationships compared to the current classification in derived Heteromantodea. 

Herein, a mitogenome-based phylogeny of Hierodulinae combined with the evidence of male genital morphology was presented. The result strongly supported the Oriental *Hierodula* and *Rhombodera* complex divided into three monophyletic groups. A new species of *Rhombodera* is herein described, and it represents the first species with leaf-like pronotum in the *Hierodula chinensis* group (species related to *H. chinensis*). Our results help to illuminate the origin and evolution of foliaceous morphology in Hierodulinae pronotum. The taxonomic implications based on the results are also discussed in regard to the future revision of the *Hierodula* + *Rhombodera* complex.

## 2. Materials and Methods

### 2.1. Taxon Sampling

Our phylogenetic analysis included 46 taxa of Mantidea *s.* Schwarz and Roy, 2019 (Mantoidea + Hymenopoidea clade) as ingroups: Mantidae (Mantinae, Tenoderinae, Hierodulinae, and Vatinae) and Hymenopodidae (Appendix A). The outgroup taxa were included as three representative species of Eremiaphiloidea based on the stable sister group relationship of Eremiaphiloidea + Pareumantodea, [10]. Sixteen mitogenomes of fourteen species were identified and sequenced in this paper (Appendix A), and seven species were included for the first time in a phylogenetic study: *Rhombomantis longipennis* Wang et al., 2021, *Rhombodera kirbyi* Beier, 1952, *R. megaera* Rehn, 1904, *Hierodula jianfenglingensis* Liu et al., 2020, *H. latipennis* Brunner de Wattenwyl, 1893, *H. confusa* Vermeersch and Unnahachote, 2020 and *R. hyalina* **sp. nov.**. The present sampling represents the species of three diverse clades in the oriental *Hierodula*+*Rhombodera* complex [12,21]: *Hierodula membranacea* group, *Hierodula patellifera* groups, and genus *Rhombodera*. 

The specimens were collected in both daytime and night during 1982–2021. Nymphal individuals at the time of collection were reared for obtaining the adult and ootheca specimens within an RDN intelligent climatic chamber with the environmental parameters following settings: temperature (25 °C); relative humidity was (70 ± 5)% and photoperiods were L/D = 18/6 h with light 9000 lx. Adult individuals were euthanized with ethyl acetate and stored temporarily in 75% alcohol for examination. Male terminalia was prepared by macerating the apical portion of the abdomen in hot 20% KOH for 10 min to isolate the genital structures. Detailed external features and male genitalia were examined based on dry pinned specimens under an OPTEC (Chongqing, China) stereoscope and captured the image information using the corresponding OPT pro application. Other photographs were taken using a Canon EOS700D digital camera. All photographs were processed using Adobe Photoshop CC2017. Morphological terminology and abbreviation follow [10,12,22]. Measurements are in mm. All of the samples were preserved at the Entomological Museum of China Agricultural University (CAU).

### 2.2. DNA Extraction and Mitogenome Sequencing

Total genomic DNA was extracted from the thoracic and coxal muscle tissue of a single specimen using a DNeasy Blood and Tissue Kit (Qiagen, Hilden, Germany). Sixteen Illumina TruSeq libraries with 450 bp average insert size were prepared and sequenced using the Illumina Hiseq2500 platform with 250 bp paired-end reads. Raw reads were trimmed of adapters and then the low-quality and short reads were removed with PRINSEQ [23]. High-quality reads were used for de novo assembly using IDBA-UD [24] with minimum and maximum k-values of 60 and 200 bp, respectively. Gene sequences of mitogenomes were annotated using Mito Z [25], and further corrected in Geneious v.10.1.3 [26] by aligning the sequences with those of homologous genes of other Mantodea species.

### 2.3. Phylogenetic Analysis

Each PCG sequence of mitochondrial DNA was aligned based by using codon-based multiple alignments under the MAFFT algorithm [27] on TranslatorX [28] online platform with the L-INS-i strategy. Sequences from two rRNA genes were separately aligned using the MAFFT v.7.0 online server with G-INS-i strategy [29]. MEGA v.6.0 [30] was used to check, concatenate, and output these alignments. Three datasets were finally outputted: (i) all 13PCGs and two rRNA genes (PCGRNA; 12754 bp); (ii) all 13 PCGs with third codon positions removed and two rRNA genes (PCG12RNA; 9171 bp), and all 13 PCGs translated to protein (AA; 3583 amino acids). The dataset PCGRNA failed to recover the monophyly of Mantidae under the best-fit GTR+F+I+G4 model (Appendix A), thus was not included in the further analyses.

Two partitioning schemes (gene-partition and non-partition) were used for AA and PCG12RNA datasets to construct the phylogenetic trees under maximum likelihood (ML) methods and Bayesian inference (BI). For ML analyses, the optimal partitioning scheme and best substitution model for each partition were selected under the ModelFinder [31] in IQ-TREE 1.6.12 [32]. The “greedy” algorithm with branch length estimated as “unlinked” was set and the AIC, AICc, and BIC criteria were chosen. ML phylogenetic trees were constructed using the IQ-TREE webserver by an ultrafast bootstrap approximation approach with 1000 replicates with the best model estimated. Bayesian analyses were carried out using MrBayes v.3.2.77 [33]. Two simultaneous runs were conducted for the datasets, and trees were sampled every 1000 generations with the first 25% discarded as burn-in. The average standard deviation of split frequencies was thought to be stationary until below 0.01.

## 3. Results

### 3.1. Taxonomy


**Subfamily Hierodulinae Brunner de Wattenwyl, 1893**



**Tribe Hierodulini Brunner de Wattenwyl, 1893**



**Genus *Rhombodera* Burmeister, 1838**


*Rhombodera* Burmerister, 1838: 536. Type species: *Mantis* (*Rhombodera*) *valida* Burmeister, 1838: 536.

**Remarks.***Rhombodera* is currently known as non-monophyletic [10,11]. Our phylogeny shows strong signal clustering of *H. latipennis*, *H. jianfenglingensis*, *H. chinensis*, *R. kirbyi*, *Rhombodera stalii* Giglio-Tos, 1912, *Rhombodera latipronotum* Zhang, 1990, *Rhombodera valida* Burmeister, 1838, *Rhombodera zhangi* (Wang and Dong, 1993), *Rhombodera megaera*, *Rhombodera hyalina* **sp. nov.**, and *R. longa* in one clade, recovering *Rhomobodera* as paraphyletic with respect to *Hierodula*. Species in this clade also share the similarity in robust pronotum of medium length with prozone 2.2–3.5 times as long as metazone, well-developed secondary distal process (sdp) in ventral phallomere not beyond the width of the main lobe of sclerite L4A, large afa in left phallomere mostly bilobed as two apical sharp lobes and elongate nearly parallel (Figure 1G–R). These characters greatly meet the typical *Rhombodera* species rather than *Hierodula* species, indicating the problematic taxonomic position of *H. latipennis*, *H. jianfenglingensis*, and *H. chinensis* currently. A large number of Southeast Asia lineages without strongly dilated pronotum of *Hierodula* are also facing such a situation. Thorough revision and redefinition of the genera in *Hierodula* + *Rhombodera* complex are urgently needed to solve the above problem.


***Rhombodera hyalina* sp. nov.**


**Type material.** Holotype (Figure 2A,B and Figure 3A,B,E) 1♂, CHINA, Guangxi, Fangchenggang, Shangsi County, 107°56′17″ E, 21°52′39.7″ N, alt. 700 m, 17.xii.2020, leg. Local collector. 

Paratypes (Figure 2C,D, Figure 3C,D and Figure 4) 2♀, CHINA, Guangxi, Qinzhou, Qinbei County, Dazhi Town, 108° E, 21° N, alt. 800 m, v.2022, local collector; 1♂, same as holotype; 1♂, China, Guangxi, Chongzhuo, Longzhou Country, Xiangshui Town, 106° E, 22° N, alt. 600 m, 18.v.1982, leg. Chi-Kun Yang.

**Diagnosis.** This new species is very similar to *R. latipronotum* and *R. valida* (Appendix A) but could be recognized from the latter two by the following features: (1) margins of the pronotum of the female slightly rippled dorsally and darkly pigmented when alive while the latter two are completely smooth and nearly unicolor dorsally (Figure 2C and Figure 4A). (2) Forewing discoidal area completely hyaline in males. (3) Narrow pterostigma of both sexes. (4) Slender and long sdp and the possession of median secondary distal process (sdpm) in the ventral phallomere (Figure 3E). (5) The mature nymphs of the new species show a ventrally upturned abdomen (Figure 5D), while it is outstretched in *R. latipronotum* (Figure 5E) and *R. valida*. The new species is also similar to Indochinese *Rhombodera* species *R. megaera* and *R. zhangi* but could be immediately distinguished by more dilated pronotum: the ratio length/width are about 1.4, 2.7, and 2.4 times of *R. hyalina*
**sp. nov.**, *R. megaera*, and *R. zhangi*, respectively.

**Description. Measurements.** See Table 1. 

**Male** (Figure 2A,B, Figure 3A,B,E and Figure 4C–F). Large-sized. Generally light green. ***Head.*** Triangular, about 0.85 times as long as wide, entirely green with drop-like large compound eyes. Vertex completely flat. Three ocelli prominent, closely approaching each other. Ocellar tubercle slightly protruding and connecting ocelli. Lower frons pentagonal, the width slightly greater than the height. A pair of feebly expressed carinae of parallel arrangement in the middle of the lower frons. Clypeus trapezoidal. ***Pronotum.*** Robust, 1.42 times as long as wide, and rhombic in dorsal view. Metazone is about 2.21 times as long as prozone. Margins and their surface entirely smooth. Generally green with two dark spots at both sides of the base, with edges of lateral pronotal expansion darkened. Lateral pronotal expansion foliaceous, strongly dilated, reaching the widest point slightly before the middle of its length. Anterior and posterior margins of pronotum on the same side form an angle of about 110°. Postcervical plate transversal saddle-shaped. ***Foreleg.*** Forecoxa with 6–8 triangular dorsal spines. Trochanter ventrally with a light brown spot at anterior tip. Coxal lobes round, asymmetric, with the upper one more developed. Forefemur long, green in dorsal and light yellow in ventral, with four posteroventral femoral spines (PvFS), four discoidal spines (DS), and 14–15 anteroventral femoral spines (AvFS). AvFS unequal in size, arranged as iIiIiIiIiIiIiiI. The 2nd, 3rd, 5th, 7th, 9th, and 14th or 2nd, 4th, 6th, 8th, and 15th AvFS as well as their base dark brown. Femoral brush ellipse-shaped, starting from the 11th and ending distally of the last AvS. Foretibia smooth, with 13–14 AvS gradually elongated and 10–11 PvS. Genicular spine very tiny. ***Midleg and hindleg.*** Slender. Midleg light green, while hindleg brownish green. Both of them without dilatations. ***Wings.*** Both forewing and hindwing well-developed, surpassing the end of the abdomen. Forewin narrow, 4.15 times as long as its width. Costal area opaque, light green in color, while the discoidal area of the forewings completely hyaline. Pterostigma narrow, light yellow, adjacent at 1/4 of the base of radius. Hindwing hyaline, triangular, the length is 2.1 times the width. ***Abdomen.*** Long, fusiform. Tan colored in the dorsal view and light green in ventral view. Tergite X transversal, arcuate. ***Male genitalia*.** Left phallomere: Sclerite L4B spoon-shaped. Sclerite L2 long, fusiform in shape, slowly narrowing towards the distal end, and suddenly contracting near the tip. Apical process paa long, strongly curved near its end, and forms a hook-like apex. Phalloid apophysis of the sclerite L1 well developed, with the lobe of the anterior one (aafa) short pencil-tip shaped, while the posterior one (pafa) long and robust. Both the two lobes of phalloid apophysis have a sharp tip. Ventral phallomere: broad, roughly square-shaped. Left edge of sclerite L4A weakly sclerotized. Primary distal process (pda) membraneous, rounded, and prominent. Two distal processes of ventral phallomere: lateral secondary distal process (sdpl) much sclerotized, long sickle-like, its length approximately equals to the width of sclerite L4A; sdpm less sclerotized, much slender, nearly 1/3 length of sdpl. Right phallomere: main posterior lobe nearly triangular, ridge-like projection (pva) and tooth-like projection (pia) smoothly connect in the shape of claws, upward-curved. 

**Female** (Figure 2C,D and Figure 3C,D). Large-sized, light green. ***Head.*** Triangular, about 1.15 times as long as wide. Vertex convex, with three small ocelli in a triangular formation. Lower frons width nearly equal to height, pentagonal with arched superior margins. ***Pronotum*** very robust, roughly regular rhomboidal in shape, 1.3 times as long as wide. Anterior and posterior margins of pronotum on the same side form an angle of about 92°. Postcervical plate transversal saddle-shaped. Margins of pronotum slightly uprolled dorsally and darkly pigmented when alive. Metazone about 2.21 times as long as prozone. ***Forelegs.*** Similar to male but much more robust, with 6–8 dorsal spines. Coxal lobes round, nearly symmetric. Trochanter unicolor in ventral view. Femoral brush ellipse-shaped, starting from 12th and ending just a little distally of 15th AvS. Spinal formula arranging as F = 4DS/15AvS/4PvS; T = 14AvS/11PvS. ***Wings.*** Well developed. Forewing leaf-like, green in costal area, partly hyaline in discoidal area, and gradually greening to anterior margin, about 2.5 times as long as wide. Pterostigma grayish white, very tiny and narrow. Hindwing hyaline, triangular, length 1.75 times width. ***Abdomen*** elongated oval-form, much more robust than male, uniformly green. 

**Ootheca** (Figure 5F). Barrel-like, with no extension at the residual process. Generally russet brown in color with thick beige external coating outside the external wall. The proximal end of the ootheca encircles the substrate completely, resulting in the ventral surface being perpendicular to the substrate. Flexible flaps in the emergence area reduced. 

**Etymology.** The name refers to the completely hyaline discoidal area in the forewing of the male adults of the new species.

**Biology.** This species has one generation per year in Guangxi. They overwinter as a diapause stage (antepenultimate instar) in November to February, mature in April to early May, and lay oothecae in early June. Some nymphs also have dark markings on their forefemur which disappear after their final molt. 

**Distribution.** China (Guangxi).

### 3.2. Phylogenetic Analysis

All topologies under ML and BI methods by PCG12RNA and AA datasets with different partition schemes successfully recovered the sister group of Mantidae and Hymenopoididae with high nodal support (BP ≥ 75%, BPP ≥ 90%) (Appendix A). In Mantidae, the four subfamilies included in our analysis, i.e., Mantinae, Tenoderinae, Vatinae, and Hierodulinae were respectively, recovered as a monophyletic clade, while their relationships were slightly different among some results from different datasets: the relationship (Mantinae + (Tenoderinae + (Hierodulinae + Vatinae))) was recovered only under our BI analysis of PCG12RNA-gene partition (Appendix A). In contrast, the alternative hypothesis (Mantinae + (Hierodulinae + (Vatinae + Tenoderinae))) was supported in our AA datasets by BI and ML analyses as well as PCG12RNA-no partition by ML. 

In Hierodulinae, our analyses support the monotypic genus *Mekongomantis*, with most of the results (with the exception of PCG12RNA-no partition by BI and AA-gene partition by ML placing the genus at the base of *Rhomobodera*) placing the genus as the sister group of *Hierodula* + *Rhombodera* complex with high nodal support (BP = BPP = 100%), which is consistent with Shi and Yuan and Shi et al. [20,34] The rest of the species were divided into three main lineages: *membranacea* group (*H. membranacea* and *H. confusa*), *patellifera* group (*Rhombomantis fusca*, *R. longipennis*, and *H. patellifera*), and *Rhombodera* group (*R. kirbyi*, *R. stalii*, *R. latipronotum*, *R. valida*, *R. zhangi*, *R. megaera*, *H. latipennis*, *H. chinensis*, *H. jianfenglingensis*, *R. longa*, and *R. hyalina* **sp. nov.**). *Rhombodera* group was recovered to be sister group to *membranacea* group + *patellifera* group by AA-no partition (BPP = 1), PCG12RNA-gene partition (BPP = 92%) under BI as well as AA-no partition (BP = 42%) and PCGRNA-no partition (BP = 61%) under ML. The different topology, i.e., *membranacea* group + (*patellifera* group + *Rhombodera* group), was recovered by PCG12RNA-gene/no partition, but with low nodal support (BP = 57%, 67%). The genus *Rhombomantis* was not recovered as a monophyletic group in AA analysis under ML and BI as well as in PCG12RNA-no partition under BI, and the genus *Rhombodera* was recovered as paraphyletic with respect to *Hierodula* in all analyses.

### 3.3. General Features of Newly Sequenced Mitogenomes

A total of sixteen complete or near-complete mitogenomes were assembled in this study. The length of them ranged from 17298 bp in *R. valida* to 15391 bp in *H. latipennis table* (Table 2). All of the sequences share the identical gene arrangement to the putative ancestral insect. The mitogenomes of *H. chinensis*, *H. confusa*, *H. membranacea*, *H. jianfenglingensis*, and *R. longipennis* contained 13 protein-coding genes (PCGs), 22 transfer RNA genes (tRNAs), two ribosomal RNA genes (rRNAs), and a control region. Pseudogene *trnM** was found at the 3′ end of a large noncoding region (LNCs) between *trnM* and *nad2* in *R. longa*, *R. valida*, *R. kirbyi*, *R. latipronotum*, *R. hyalina*
**sp. nov.**, and *R. Stalli*, which was also reported in seven other Hierodulinae species [35]. For those nearly complete mitogenomes, the *trnI*, *trnQ,* and *trnM* were not found in *H. latipennis*, and the *trnI* and *trnQ* were not found in *R. zhangi*, and *R. megaera*. A/T biased of mitogenomic nucleotide composition was also found in all sixteen newly sequenced Hierodulinae mitogenomes, ranging from 76.20% in *R. kirbyi* to 74.24% in *R. valida* from Java. In addition, the mitogenomes had an overall positive AT-skew (0.04 to 0.06) and negative GC-skew (−0.24 to −0.19) (Table 2).

## 4. Discussion

### 4.1. Taxonomic Implication for the Future Revision

The phylogenetic trees (Figure 6 and Appendix A) clearly divided the “*Hierodula* + *Rhombodera*” complex into three main clades: the clade “*membranacea* group”, which includes type species of *Hierodula*, *H. membranacea*. This clade shares similarities of the slender pronotum, more than ten fine spinules arranging along the dorsal coxa of the forelegs, the punctiform pterostigma, the intersegmental coloration of the ventral abdomen, the ear-like phalloid apophysis (afa), and the extremely elongated sdpl as well as its base (Figure 1B,C). *Rhombodera extensicollis* is also inferred to be part of this group because of the presence of the above characters [12,36], although still lacking molecular evidence currently. The “*patellifera* group” is represented by the species related to *H. patellifera* and the genus *Rhombomantis*. Similarities of this group are the relatively stubby pronotum, the simplified and not bilobed afa, and the long spear-shaped sdpl with its length slight beyond the width of the main lobe of the sclerite L4A (Figure 1B–F). The clade with *H. latipennis* at the base (*Rhombodera* group) is a highly diverse group, including species both with and without lateral pronotal expansion. The similarities of this clade include the medium-length pronotum, the well-developed secondary distal processes, and the distinctly bilobed afa (anterior lobe, aafa, and posterior lobe, pafa) (Figure 1G–R). At least one of the two lobes in *Rhombodera* group possesses a sharp apex; there is also a parallel tendency in the directions of the elongating of the two lobes in this group (vs. deeply forked afa in Tenoderinae and Vatinae). Combined with the above morphological and molecular evidence, the presence of well-developed lateral expansion should not be homologous in many species, and at least some species with weak lateral expansion might be very closely related to *Rhombodera*, rendering the well-developed lateral expansion invalid as the diagnosis of *Rhombodera*. 

Two treatments could be applied to resolve the unnatural grouping of *Rhombodera*. One of them could erect new taxa or remove described taxa from synonyms to accommodate several subclades in the *Rhombodera* group, while male genitalia will be the main diagnosis of these taxa. The alternative treatment could be the rearrangement of the hieroduline species between existed genera in precedence. It may require a large number of species to be transferred from *Hierodula* to *Rhombodera* as well as expanding the definition of *Rhombodera*, and finally, *Rhombodera* will at least include the *Rhombodera valida* group and *Hierodula chinensis* group from the oriental realm. Our results are here inclined to support the second one in dealing with the unnatural grouping in the complex. Such treatment provides higher stability and fault tolerance, and the diagnosis of the three main clade of this complex will be quite simple, though the final results of this way may cause a great increase in the species number of *Rhombodera*. The further division of *Rhombodera* into several genera would be over-reliant on male genitalia because almost no other stable morphological characters could be concluded, which might reduce the practicality at the genus level. It is also worth noting that some synonymies of hieroduline taxa were established based on problematic species itself, and recovering the validity of these taxa is also an urgent problem.

Our study did not include the Australasian and Papua lineages such as *Tamolanica* and the species related to *Hierodula majuscula* of the *Hierodula*+*Rhombodera* complex. However, we conclude that it will not influence the validity of the above discussion because the ancestral position of these lineages [35] and distinct male genitalia [10] give this lineage relative independence compared to the oriental lineages.

### 4.2. Evolutionary Significance of the New Species

Our findings indicate that *Rhombodera* lineage also has species with the narrow pronotum, and the new species presents foliaceous pronotum again in a sub-clade without such character. Combining this new finding with our phylogeny, the pronotal foliaceous mimicry was observed in at least five lineages in oriental hieroduline species (Figure 6), suggesting a complex evolutionary pattern of pronotal morphology. The repeating appearance of leaf-like pronotum in multiple clades justifies the particularity and adaptive significance of this phenomenon in Mantodea. 

## Figures and Tables

**Figure 1 insects-13-00715-f001:**
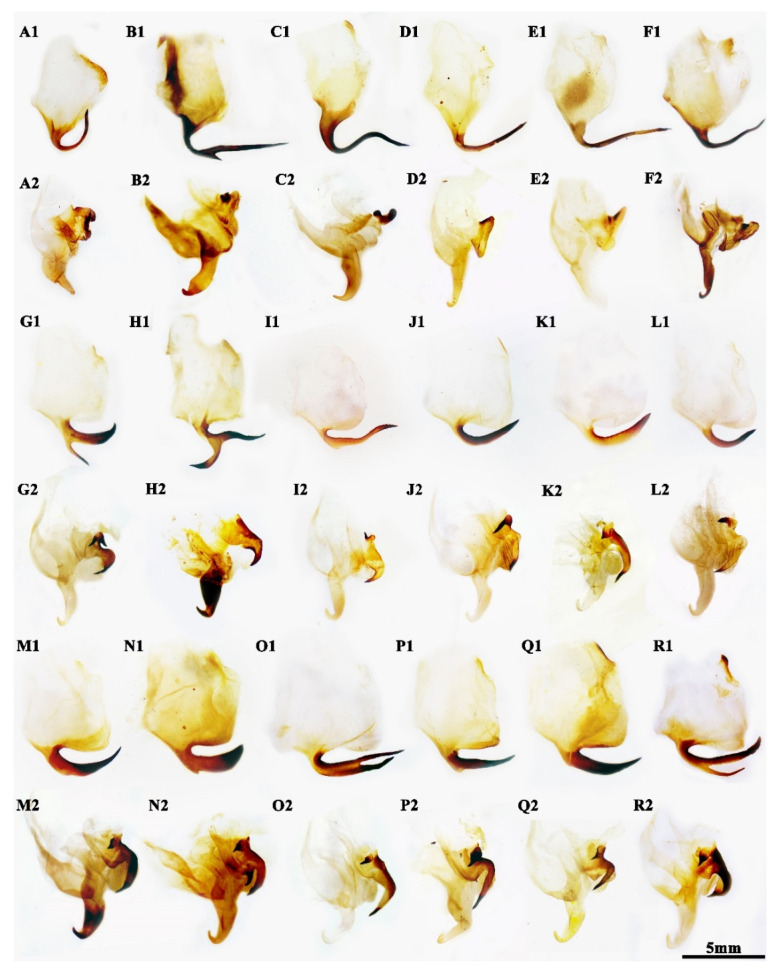
Male genitalia of *Hierodula* and *Rhombodera* ((**1**) ventral phallomere; (**2**) left phallomere; Right phallomeres omitted). (**A**) *Mekongomantis quinquespinosa*. (**B**) *Hierodula confusa*. (**C**) *Hierodula membranacea*. (**D**) *Hierodula patellifera*. (**E**) *Rhombomantis longipennis*. (**F**) *Rhombomantis fusca*. (**G**) *Hierodula latipennis*. (**H**) *Rhombodera kirbyi*. (**I**) *Rhombodera stalii*. (**J**) *Rhombodera latipronotum*. (**K**) *Rhombodera valida* (Java). (**L**) *Rhombodera valida* (Yunnan). (**M**) *Hierodula jianfenglingensis.* (**N**) *Rhombodera zhangi*. (**O**) *Rhombodera megaera*. (**P**) *Hierodula chinensis*. (**Q**) *Rhombodera longa*. (**R**) *Rhombodera hyalina*
**sp. nov.**.

**Figure 2 insects-13-00715-f002:**
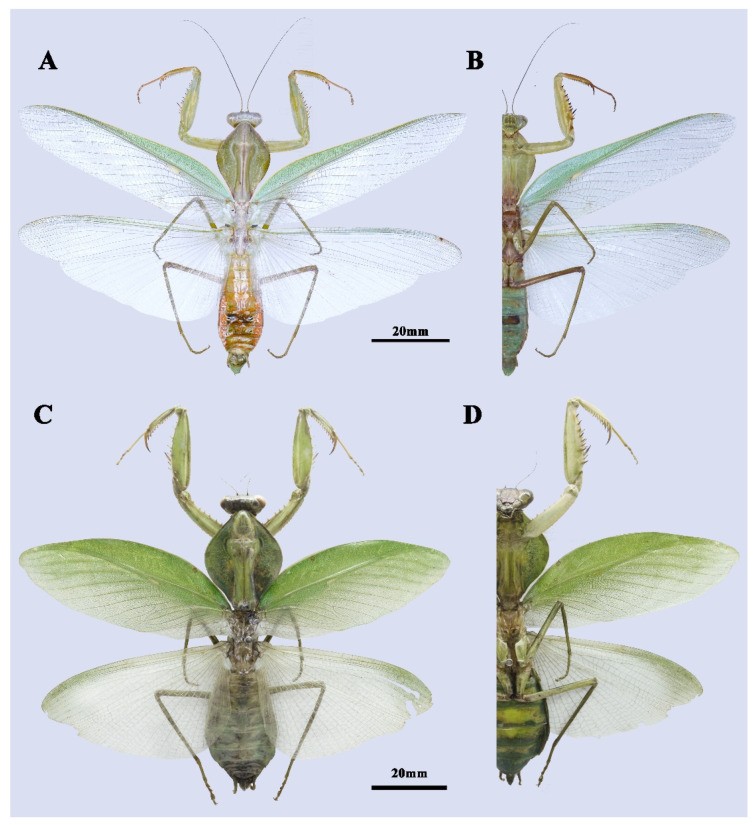
*Rhombodera hyalina* **sp. nov.** (**A**,**B**) Holotype, male. (**A**) Dorsal view. (**B**) Ventral view. (**C**,**D**) Paratype1, female. (**C**) Dorsal view. (**D**) Ventral view.

**Figure 3 insects-13-00715-f003:**
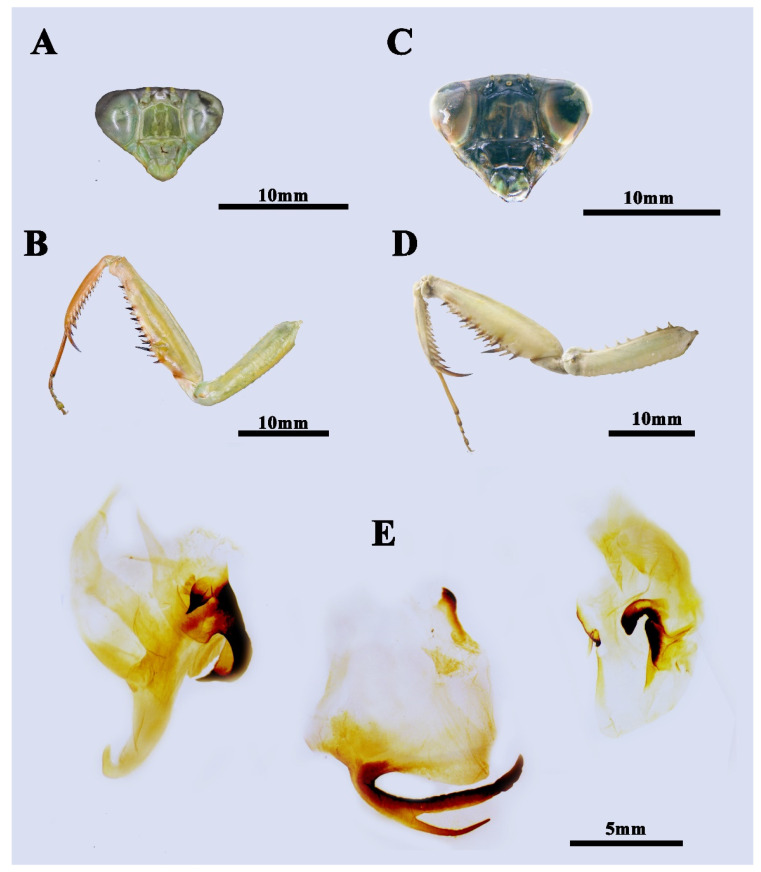
The anatomical structures of *Rhombodera hyalina* **sp. nov.** (**A**,**B**) Holotype, male. (**A**) Frontal view of the head. (**B**) Ventral view of forelegs. (**C**,**D**) Paratype1, female. (**C**) Frontal view of the head. (**D**) Ventral view of forelegs. (**E**) Male genitalia. Left: left phallomere; the middle: ventral phallomere; Right phallomere.

**Figure 4 insects-13-00715-f004:**
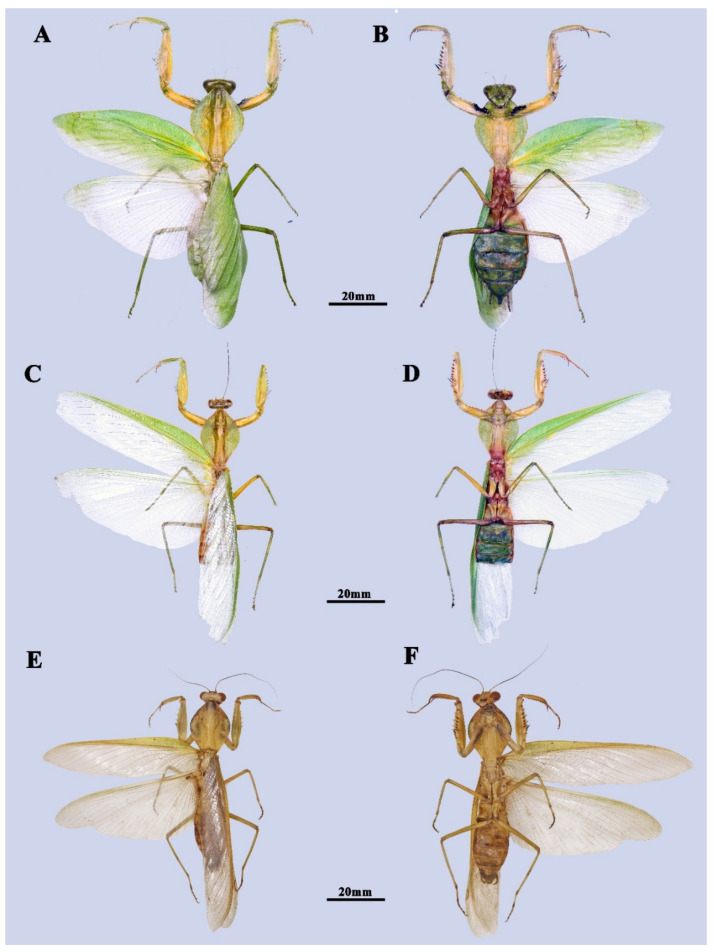
*Rhombodera hyalina* **sp. nov.**, paratype. (**A**,**B**) Paratype2, female. (**C**,**D**) Paratype3, male. (**E**,**F**) Paratype4, male.

**Figure 5 insects-13-00715-f005:**
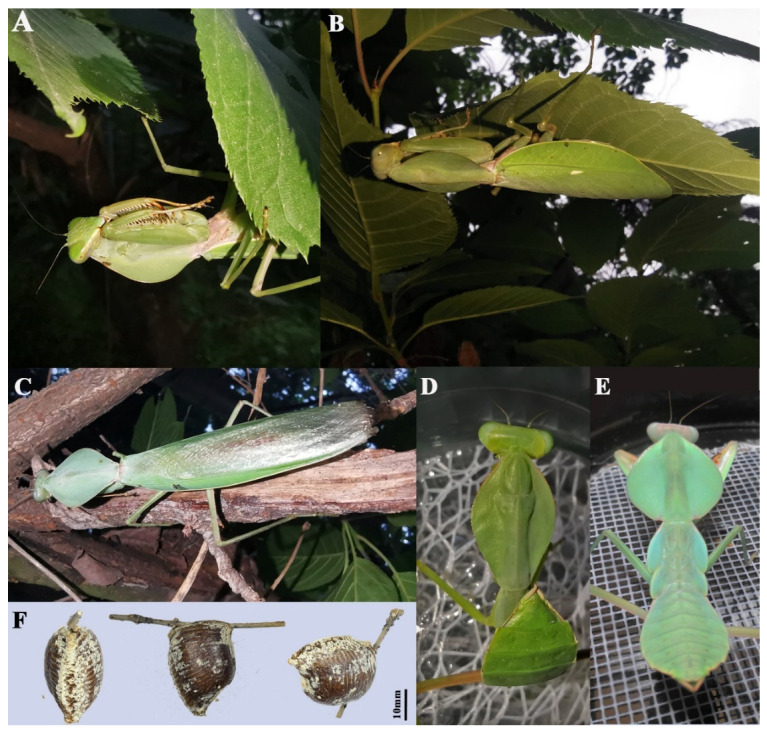
*Rhombodera hyalina***sp. nov.**, ecological details. (**A**,**B**) A living female. (**C**) A living male. (**D**,**E**) Comparison of the older nymphs of *R. hyalina* (on the left) and *R. latipronotum* (on the right). (**F**) Ootheca of *Rhombodera hyalina*.

**Figure 6 insects-13-00715-f006:**
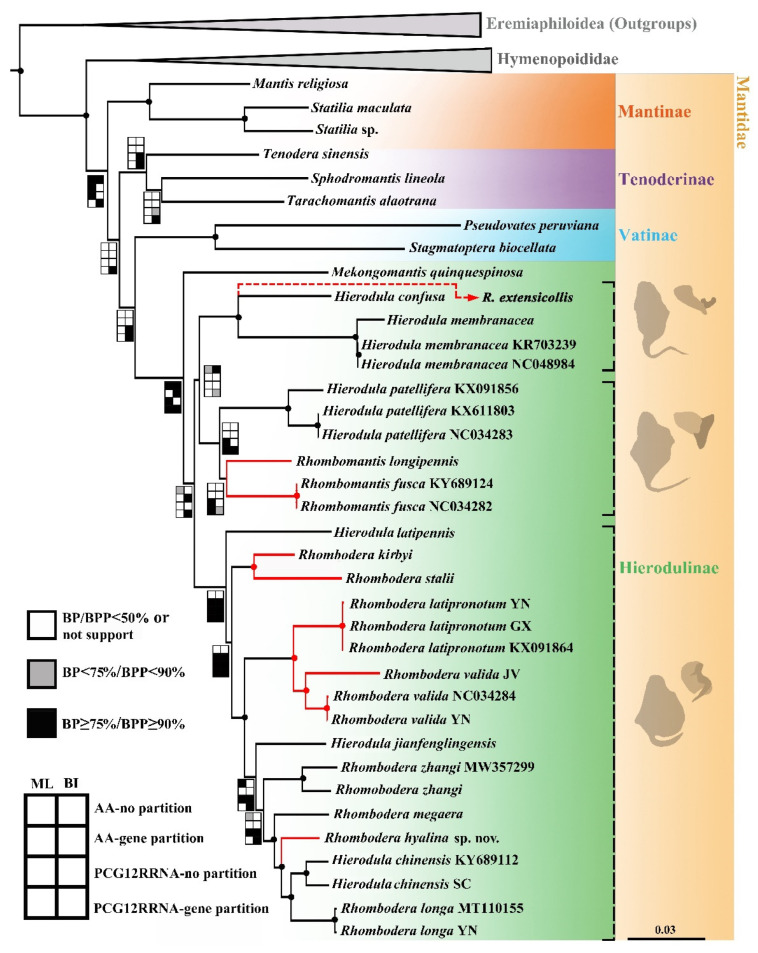
Phylogeny of Hierodulinae from PCG12RNA-gene partition under BI. The relationships that are well supported in all eight analyses are marked in a circle at the nodes. Red full line clades indicate pronotal foliaceous mimicry lineage included in our phylogeny, while the red dashed line indicates the putative position of *R. extensicollis* according to the morphology. The silhouettes in the yellow area are derived from the typical ventral phallomere (the left) and afa on the male genitalia of three *Hierodula* + *Rhomobodera* groups. Abbreviations of the location: JV, Indonesia, Java; YN, China, Yunnan; SC, China, Sichuan. The operational taxonomic units without an accession number followed are generated by this study.

**Table 1 insects-13-00715-t001:** Detailed measurements (in mm) of *Rhombodera hyalina* **sp. nov.**.

Measurements	Holotype ♂	Paratype1 ♀	Paratype2 ♀	Paratype3 ♂	Paratype4 ♂
Total length	80.16	82.88	—	r81.00	84.42
Body length	61.86	73.60	74.72	61.76	71.36
Head width	9.72	11.23	11.53	9,52	9.68
Head height	7.36	9.90	9.80	7.76	7.56
Pronotum length	19.94	24.58	25.50	20.50	21.36
Pronotum width	14.00	18.10	19.37	13.66	15.28
Pronotum narrow width	4.42	4.66	4.98	4.80	4.66
Prozone length	6.22	7.19	7.70	6.30	6.30
Metazone length	13.72	17.54	17.80	14.20	15.06
Forewing length	60.34	55.50	53.88	59.38	62.38
Hindwing length	54.58	49.14	51.28	54.50	56.54
Forecoxa length	13.92	18.34	17.39	13.66	14.80
Forefemur length	15.10	19.12	19.99	15.32	16.38
Foretibia length	10.58	11.78	12.18	10.76	11.52
Foretarsus length	10.34	12.00	11.90	10.60	12.48
Mesofemur length	14.22	16.54	15.64	14.46	15.18
Mesotibia length	11.00	14.42	13.60	11.14	12.14
Mesotarsus length	8.18	8.78	7.30	8.50	8.68
Metafemur length	15.32	19.56	17.98	16.08	17.76
Metatibia length	16.12	20.72	17.74	16.36	18.82
Metatarsus length	11.16	11.26	10.90	11.18	11.82

**Table 2 insects-13-00715-t002:** Base composition of sixteen Hierodulinae mitogenomes.

Species	A + T (%)	AT-Skew	GC-Skew	Total Length
*Hierodula chinensis* (Sichuan)	75.80	0.06	−0.21	16,145
*Hierodula confusa*	75.00	0.05	−0.23	16,103
*Rhombodera longa*	75.90	0.06	−0.21	16,093
*Hierodula membranacea*	74.87	0.05	−0.23	15,824
*Hierodula jianfenglingensis*	75.32	0.05	−0.20	16,234
*Hierodula latipennis*	75.91	0.05	−0.20	15,391
*Rhombodera zhangi*	75.58	0.06	−0.22	15,396
*Rhombodera valida* (Java)	74.24	0.05	−0.19	17,298
*Rhombodera kirbyi*	76.20	0.05	−0.20	16,036
*Rhombodera latipronotum* (Yunnan)	75.10	0.06	−0.22	16,092
*Rhombodera latipronotum* (Guangxi)	75.05	0.06	−0.23	16,168
*Rhombomantis longipennis*	75.70	0.04	−0.21	16,056
*Rhombodera megaera*	75.33	0.06	−0.20	15,875
*Rhombodera hyalina*	75.69	0.05	−0.22	16,043
*Rhombodera stalii*	75.33	0.06	−0.22	16,314
*Rhombodera valida* (Yunnan)	75.18	0.06	−0.24	16,055

## Data Availability

The data supporting the findings of this study are openly available from the National Center for Biotechnology Information at https://www.ncbi.nlm.nih.gov (accessed on 1 August 2022). Accession numbers are: OP108446, OP168273, OP168274, OP168275, OP168276, OP168284, OP168278, OP168277, OP168279, OP168283, OP168287, OP168286, OP168280, OP168282, OP168281, OP168285.

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
