# Peer review of "Mitochondrial Phylogenomics Suggests Complex Evolutionary Pattern of Pronotal Foliaceous Mimicry in Hierodulinae (Mantodea: Mantidae), with Description of a New Species of Rhombodera Burmeister, 1838 from China†"

_insects, 2022, doi:10.3390/insects13080715_

Round 1

Reviewer 1 Report

This paper describes a new mantid species (genus Rhombodera) and uses mitochondrial genomes to assess relationships within the Hierodulinae.

The species description seems thorough (but I am not a taxonomist and cannot really comment here). The mitochondrial genome part/phylogenetic analysis is hard to follow. Although the methods are explained in detail and the approach followed is sound, it is difficult to get an overview of the species included, the newly sequenced ones and their genome characteristics (the results section is very brief on this). A lot of (important) information seems to be placed in the supplement. As 'Insects' has no page limit it is unclear why the authors decided to do this - I suggest the authors to move supplementary material (tables) to the main document. This will help the reader/improve the clarity of the paper. Another option might be to split the paper into two separate ones: one focussing on the species description, the second focussing on the phylogenetic analyses and reclassification. 

Minor comments:

Summary

"according to the results above" --> remove as no detailed/specific results were presented 'above'.

"Our results show that the independent origin in multiple lineages of foliate pronotum in Hierodulinae may have adaptative significance" --> how can a phylogenetic tree show this? Please rephrase. Also adaptative is incorrect.

"...male genitalia..." --> and vice versa? that male genitalia provide good characters for determining relationships within the group?

"three monophyletic clades" --> clades are monophyletic by definition. Change sentence. 

Introduction:

"while few have actually solved it." --> So relatioships have actually already been resolved? If not, change sentence.

Results:

"were slightly different among some results from different datasets" --> very vage. Please rephrase

Figure 5: Please highlight on the figure which sequences were generated by you and add Accesssion numbers for these.

Discussion:

"also possesses the foliaceous pronotum lineage" --> it does not possess this lineage. It contains a species that shows a foliaceous pronotum. 

"proved that the pronotal foliaceous mimicry happened at least five times" --> replace 'proved' with 'indicates' or 'suggests'.

Also, your tree could also suggests it evolved 4 times and was lost in the clade that has R. jianfenglingensis at the base. This would be a more parsimonious explanation. 

" and adaptative significance of this phenomenon. " --> I don't understand this. How can you conclude (from not all species having the trait) that it is 'adaptive'  (not adaptative).

The discussion ends abruptly. Can you add a final take home message? what is the key message of the paper?

Reviewer 2 Report

Liu et al. provide a mitochondrial phylogeny of a subset of taxa from the hyperdiverse Hierodulinae, describe a new species, and make an attempt at transferring some species from Hierodula to Rhombodera as a consequence of their results.

The paper is generally well written, the English is much better than previous papers of these authors I got for review. Only minor spelling corrections were necessary.

Figures 1 and 2 should be separeted into multiple figures, showing anatomical and genital traits at larger size. As they are now, not much detail can be discerned.

The new species seems justified to me (also here, the genitalia should be presented at higher resolution), also the generic placement. But, to be sure, I suggest the authors compare their species with all Chinese and Indochinese Rhombodera via a differential diagnosis (there is one already, but I suggest to expand it beyond the two species mentioned). The female is rather succinctly described. This is more usual in already described species, but a new taxon should be described in as much detail as possible without being redundant, particularly all characters needed for identification and distinction from related taxa should be clearly worked out.

I see a major issue with the transfer of H. latipennis, H. chinensis and H. jianfenglingensis to Rhombodera. For reasons given in the comments to the attached file, this would not create an improvement over the current situation, but just the movement from one wastebasket taxon to another. A clade is not necessarily identical to a genus, and the ommission of most hieroduline genera and of most of SE Asian, Wallacean and Australo-Papuan members of Hierodula from the phylogeny does not allow any conclusions yet on the number of genera needed to properly describe the monophyletic units of the subfamily. I recommend the authors refrain from this step and leave the species where they are for now. Only a revsion of the whole subfamily will justify any steps of this kind. For more explanations see attached file.

I recommend this paper for publication after a major revision.

Round 2

Reviewer 2 Report

The authors have commented upon my previous comments and have improved the paper accordingly, with one exception.

They still want to undertake taxonomic actions and transfer a subset of Hierodula species to Rhombodera. As explained in the attached file, the arguments presented by the authors in favor of their approach actually support my opinion to not transfer those taxa. The most problematic aspect here is that they ommitted taxa from the phylogeny, whose genitalia look different from the three groups presented here. Their addition will have disrupted the seemingly homogenous genital morphology in their groups. The same holds true for the taxa included in Wang et al. (2022), a paper referenced in support of the author's actions, but which actually contradicts it.

I cannot endorse the taxonomic actions undertaken by the authors due to lack of data supporting it, and too much available data contradicting it (see comments in the file). Exception: the description of the new species seems justified to me.

Round 3

Reviewer 2 Report

This version is good to go, but please correct a few last spelling suggestions in the attached file.
